# The influence of health facility-level access measures on modern contraceptive use in Kinshasa, DRC

**Saleh Babazadeh**[1]*, **Philip Anglewicz**[2], **Janna M. Wisniewski**[1], **Patrick K. Kayembe**[3], **Julie Hernandez**[1], **Jane T. Bertrand**[1]

**1** Department of Health Policy and Management, Tulane School of Public Health and Tropical Medicine, New Orleans, Louisiana, United States of America, **2** Department of Population, Family and Reproductive Health, Johns Hopkins Bloomberg School of Public Health, Baltimore, Maryland, United States of America, **3** Kinshasa School of Public Health, University of Kinshasa, Kinshasa, Democratic Republic of Congo

* sbabazad@tulane.edu

**Data Availability Statement:** Data can be found on PMA2020 website and is available upon request at the IPUMS PMA repository at: https://pma.ipums.org/pma/index.shtml IPUMS PMA is a publicly

## Abstract

Expanding access to family planning (FP) is a principal objective of global family planning efforts and has been a driving force of national family planning programs in recent years. Many country programs are working alongside with the international family planning community to expand access to modern contraceptives. However, there is a challenging need for measuring all aspects of access. Measuring access usually requires linking information from multiple sources (e.g., individual women and facilities). To assess the influence of access to family planning services on modern contraceptive use among women, we link four rounds of individual women and service delivery points survey data from PMA2020 in Kinshasa. Multilevel logistics regression on pooled data is performed to test the influence of facility-level access factors on individual-level contraceptive use. We add variables tailored from a conceptual framework to cover elements of access to family planning: administrative access, geographic or physical access, economic access or affordability, cognitive access, service quality, and psychological access. We find that the effect of community and facility-level access factors varies extensively but having fewer stocked-out facilities and more facilities with long-acting permanent methods (LAPM) increases the odds of using modern contraceptives among women in Kinshasa. Our study shows that reliable supply chain with a broad array of method mix will increase the odds of modern contraceptive use at community level among women in Kinshasa. Using to community-oriented practices and service delivery along with empowering women to make health-related decisions should become a priority of family planning programs and international stakeholders in the country.

## Introduction

### Background

Expanding access to family planning (FP) is a principal objective of global family planning efforts and has been a driving force of national family planning programs in recent years. Like

available data that harmonizes the international family planning survey series Performance Monitoring for Action, or PMA (formerly known as Performance Monitoring and Accountability 2020 or PMA2020)

**Funding:** BMGF is the grant awarded to Tulane University to implement PMA2020 surveys in the DRC. the funders had no role in study design, data collection and analysis, decision to publish, or preparation of the manuscript.

**Competing interests:** The authors have declared that no competing interests exist.

many other countries' programs, the Democratic Republic of Congo's family planning program is focused on increasing contraceptive use through improving access to family planning services, and is based on the assumption that greater access will lead to increased use [1]. The DRC, with a total fertility rate of 6.3, has one of the highest fertility rates in the world. With the current population growth rate (3.1), the DRC's population is projected to increase by 131.6 million by 2050 [2, 3]. The DRC has planned to increase the modern contraceptive prevalence rate (mCPR) to 19% by 2020 (7.8% as of 2014) [4], reach an additional 2.1 million modern contraceptive users with a range of modern contraceptive methods as part of the FP2020 Initiative, and improve access to family planning services for men and women in the public and private sectors [1].

Kinshasa, the capital of the DRC, plays a key role in accomplishing these objectives. The urban area of Kinshasa is populated by an estimated 12 million population in a 200 square mile area, which makes it the third-largest urban area in Africa after Lagos and Cairo [5, 6]. The mCPR among women of reproductive age in Kinshasa is higher than other provinces in the DRC (26.5% as of 2018) [7]. However, Kinshasa is yet considered to have a low prevalence compared to many other FP2020 focal countries in Sub-Saharan Africa [8].

At face value, the DRC's approach to improving access as a means of increasing utilization of FP is evidence based. Studies from the 1970s through recent years that focused on the issue of access to family planning have hypothesized that greater access would increase utilization of family planning services [9–15]. However, despite the emphasis placed on access to family planning, there is no consensus on the definition and measurement of access [16]. Many studies have used geographic access or proximity to the service delivery point (SDP) as a proxy of access [9, 10, 17–19], but access is, in fact, a more multi-faceted concept that has individual, community, policy, and facility-level elements [15, 20–23].

Quantifying such a complex concept requires multiple sources of data, including population-based surveys, facility-based surveys, and routine service statistics. Few studies have measured the magnitude of the influence of the family planning supply environment and service quality on contraceptive use, mainly because of the complexity and the lack of appropriate sources of data [15]. Linking data from those sources offers greater insight into how the availability and quality of health services in individuals' service environment can impact healthcare-seeking behavior [24]. Some studies have linked facility data from a country's Service Provision Assessment (SPA) with individual women's data from the Demographic and Health Survey (DHS) to test the influence of service availability and quality on contraceptive behavior [14, 25]. Other studies have tested this association at the cluster level [18, 26, 27]. However, none of these approaches are perfect. One weakness is the fact that the spatial data from the DHS are randomly displaced as a confidentiality measure and few SPA surveys include GPS coordinates. Another issue with such linkage is that DHS and SPA surveys are rarely executed at the same time and the same clusters within a country. As a result, there is typically a time and location difference between the individual data from DHS and facility data from SPA. Both shortcomings limits the inferences that can be made using SPA and DHS data [24, 28, 29].

The objectives of this analysis are to construct a comprehensive measure of access using variables from both population and service delivery points and to test the association of access to family planning to modern contraceptive use. Using both sources of data (population-based and facility-based), we aim to assess the individual and facility-level determinants of modern contraceptive use among all women of reproductive age (15–49 years old) in Kinshasa over the four rounds of PMA2020 surveys administered between 2014 and 2016. Specifically, we analyze the relative influence of conventional socio-demographic correlates of modern

contraceptive use (which affect the demand for FP services) and community-level factors related to the supply environment for FP services.

In this study, we investigate the following questions:

1. To what extent do EA-level FP supply and services impact modern contraceptive use among women in reproductive age in Kinshasa, DRC?

2. To what degree can the variability in contraceptive use among women in reproductive age in Kinshasa be explained by differences in EA-level contextual factor variables?

3. To what degree can the variability in contraceptive use among women in reproductive age in Kinshasa be explained by variation in other EA-level variables?

It is important to consider that the decision to utilize contraceptive services is a product of many individual, community, provider, and service level factors. Investigating these factors and their association will provide FP programs and policy makers with the necessary evidence as to the relative importance of these factors in individual women's decision to use modern contraceptives and identify the probable barriers to obtaining FP services.

## Methods

### Data

This study uses four consecutive rounds of population-based and facility-based survey data collected in Kinshasa through Performance Monitoring and Accountability 2020 (PMA2020), a multi-country platform which, similar to the DHS and SPA, surveys female respondents and service delivery points (SDP) about family planning [30]. PMA2020's SDP and population-based data are collected at the same time and in the same clusters in multiple rounds within a country. Therefore, in this study, the population and facility data are linked at the cluster level. The surveys were conducted in the same 58 enumeration areas (EA) in four annual and semi-annual rounds (rounds 2–5) between 2014 and 2016 (rounds three and four were conducted in 2015). The sample consists of a panel of 58 EA selected in four rounds of data collection. Two-stage cluster sampling was employed for the population-based survey; at the first stage of sampling, 58 enumeration areas (EA) were randomly selected from the 335 EAs within the city of Kinshasa. In the second stage, 33 households were selected within each selected EA using systematic random sampling. All women of reproductive age in each household were eligible to be interviewed.

The sampling strategy for SDPs was slightly different. At the first stage of sampling, the same 58 EAs were selected. In the second stage, the interviewers collected data from 3–6 SDPs per EA (three public and three private SDPs). The sampling approach was different for public and private SDPs. For public SDPs, the data collectors obtained a list of all public facilities, stratified by type of facility. Within each EA, the tertiary hospital that serves the EA was selected (there is only one tertiary hospital in Kinshasa). The secondary hospitals were selected if they served the selected EA, even if they were not located in that EA itself. If the EA contained primary-level public facilities (health center, health clinic, health post) that served the population in that EA, one was randomly selected.

For private facilities, resident enumerators (REs) first listed all private SDPs within each EA. Private SDPs included faith-based SDPs, pharmacies, clinics, and other unofficial providers of FP methods (such as kiosks). Three private SDPs were selected randomly from the prepared list. As a result of sampling approach, all EAs have at least three private SDPs while only some have three public SDPs. Further, some SDPs were selected for the sample in multiple rounds. However, since they are a small fraction of all SDPs (less than 8% of all SDPs), and

removing them from the analysis did not meaningfully change the results, we retained the SDPs that where selected in multiple rounds.

This study received IRB approval from Tulane SPHTM (Ref #492318) and from the Kinshasa School of Public Health (2013–2014: ESP/CE/070/13; 2015–2016: ESP/CE/070b/2015; 2015–2016: ESP/CE/070c/2015). All women and SDP staff interviewed gave informed consent and the data was fully anonymized before the analysis.

## Conceptual framework

For this study we quantified access to family planning using six elements: administrative access, geographic or physical access, economic access or affordability, cognitive access, service quality, and psychological access [15, 23]. These elements are shown in Fig 1. *Administrative access* indicates the degree to which administrative barriers are eliminated, for example, whether the facility has restricted working hours or days, restrictive policies that lead to discrimination (e.g., age or marital status restrictions), or additional requirements to serve clients (e.g., husband's approval). *Geographic access* is defined as the extent to which facilities are located so that the majority of the population can reach them with a reasonable amount of time and effort. Some consider the cost that an individual incurs to reach the facility for the FP services as a component of geographic access. *Cognitive access* is the extent to which potential users are aware of services (such as the existence of individual FP methods) and facilities for FP methods. *Psychological access* measures the extent to which social or attitudinal factors constrain potential users in seeking contraceptive methods. Examples of such factors include a husband's opposition to FP or negative attitudes or behaviors on the part of the facility staff. *Economic access* demonstrates the extent to which obtaining contraceptive methods is within the economic means of the client. In other words, it is the individual's ability and readiness to pay the service provider's fees. Finally, *service quality*

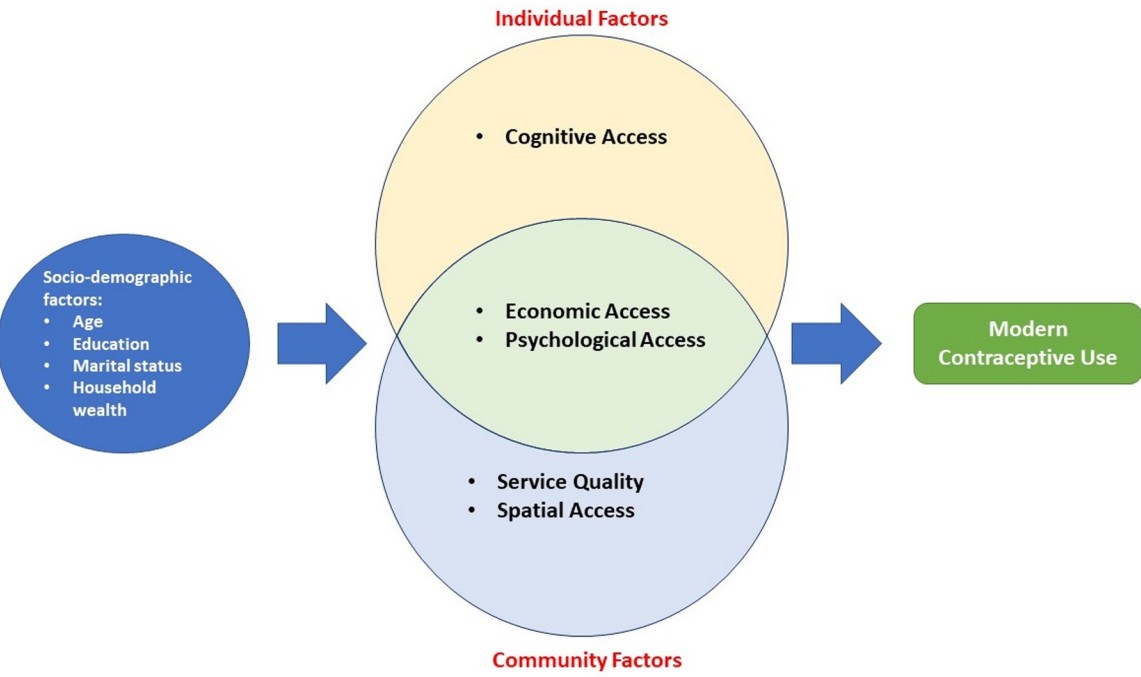

**Fig 1. The conceptual framework of access to modern contraceptive.**

refers to the availability of a range of methods and counseling on these methods; in addition, it encompasses the extent to which the facility has necessary commodities, trained staff, required equipment, and appropriate infrastructure for service delivery (e.g., a proper room for inserting an IUD) [15, 23].

## Measurement

**1. Outcome variable.** The binary outcome variable, modern contraceptive use, indicates whether a woman of reproductive age–regardless of marital status–reported using a modern contraceptive to delay or prevent pregnancy at the time of the survey. In this study, modern methods of contraception are oral pills, injectables, male or female condoms, intrauterine devices, male or female sterilization, and lactational amenorrhea. Women are not considered to be using a modern method if they reported not using any contraceptive method or using periodic abstinence, withdrawal, or other traditional family planning methods.

**2. Key independent variables.** Three groups of independent variables were tested in this analysis.

**a)** The first group of independent variables is a proxy for the availability of contraceptive methods and the supply environment within the SDPs. These variables were constructed as the aggregate of availability and quality of service variables at the EA level. For this analysis, we constructed 11 EA-level SDP variables to measure elements of access to FP. The constructed SDP variables are as follows: total number of SDPs that offer family planning methods, total number of methods per SDP, total number of SDPs with more than three methods in stock, total number of SDPs with more than five methods in stock, total number of days SDPs offer FP services, total number of long-acting and permanent methods (LAPM), total number of SDPs that have the capacity to insert implants, total number of SDPs that have the capacity to insert IUDs, total number of SDPs that have been stocked-out in the past three months, total number of SDPs with fees, and total number of SDPs with water and electricity.

**b)** The second group of independent variables measures contextual factors ("community influences") by calculating the mean values found for individuals on specific variables to create a "community-level" variable. Previous studies indicate that for contraceptive use, women usually navigate community norms to fulfill their ideas in terms of fertility and contraceptive decision- making [31]. A growing body of literature has investigated the role of these contextual factors in women's contraceptive use in Sub-Saharan Africa [14, 32–35]. The purpose of adding this set of control variables was to test if contextual, community-level variables have an impact on modern contraceptive behavior in Kinshasa. We constructed the level of knowledge of contraceptive methods (as the mean number of methods known) in each EA, mean wealth index in each EA, mean age in each EA, mean number of children in each EA, and the proportion of women making the final decision in each EA as the community-level variables.

**c)** The third group of independent variables is individual-level control variables. These variables are those have been identified as determinants of modern contraceptive use in many sub-Saharan African countries: age, household wealth index, level of educational attainment, parity, marital status, exposure to media messages on family planning (television, radio, newspaper/magazine), knowledge of contraceptive methods, and desire for more children [36, 37]. Table 1 shows the definition of all three groups of variables with the dimension of access that they represent.

**Table 1. Variables representing elements of access to family planning.**

| Variables | Definition | Element of FP access |
|---|---|---|
| **EA-level SDP variables** | | |
| Number of SDPs offer FP | Number of SDPs that offer FP services in each EA | Geographic access |
| Number of methods per EA | Number of FP methods offered in each EA | Service quality |
| Number of SDPs > 3 methods | Number of SDPs that offer at least 3 FP methods in each EA | Service quality |
| Number of SDPs > 5 methods | Number of SDPs that offer at least 5 FP methods in each EA | Service quality |
| Number of days SDPs | The average number of days per week, SDPs are open in each EA | Administrative access |
| Number of LAPM | Number of LAPM methods per SDP in each EA | Service quality |
| Number of SDPs insert implant | Number of SDPs that offer FP services in each EA | Service quality |
| Number of SDPs insert IUD | Number of SDPs that offer FP services in each EA | Service quality |
| Number of SDPs stocked-out | Number of SDPs that have been stocked out in the last 3 months on any method in each EA | Service quality |
| Number of SDPs with fees | Number of SDPs that offer FP services with fees in each EA | Economic access |
| Number of SDPs with water and electricity | Number of SDPs that offer FP services and have running water and electricity in each EA | Service quality |
| **Individual and household variables** | | |
| Age | Self-reported age of respondent at time survey | Socio-demographic |
| Level of education | Highest educational level attained | Socio-demographic |
| Parity | Number of children given birth | Socio-demographic |
| Marital status | Woman's marital status | Socio-demographic |
| Wealth index | Household wealth index in quintile | Socio-demographic /Economic access |
| Media Exposure to FP messages | Number of media exposures (radio, television and newspaper (0–3)) | Cognitive access |
| Desire for having more children | Individual wants more children as woman | Socio-demographic |
| Number of methods known | Number of FP methods the individual woman knows | Cognitive access |
| Round of PMA2020 | Round of PMA2020 data collection in Kinshasa (2–5) | |
| **EA-level community variables** | | |
| Community level of FP knowledge | Average number of methods known by women in each EA | Cognitive access |
| Community level of wealth | Average wealth index in each EA | Economic access |
| Community level of FP decision | Average percentage of women who made decision on contraceptive use on current or previous FP use | Cognitive access |
| Community level parity | Average number of births per woman in each EA | Socio-demographic |

## Analysis

Descriptive analysis was performed for both individual and EA-level variables on a pooled cross-section of data from the four rounds of PMA2020 in Kinshasa. We tabulated the demographic characteristics of the women in our study population for each round of data. We also assessed statistical differences in these characteristics across four rounds of PMA2020 and conducted a bivariate analysis to test the association of both individual and EA-level SDP variables with the modern contraceptive use across four rounds.

Finally, we examined whether modern contraceptive use differs significantly by particular EA-level characteristics of supply environment and other EA-level community variables controlling for individual variables. Multilevel logistic regression was used to estimate the effect of EA-level variables and individual variables in the outcome of interest. This method is chosen for two main reasons: first, PMA2020 population survey data is sampled in a hierarchical structure, which means individuals are nested within EAs. Thus, we assume that respondents who live in the same EAs may not be independent of one another. Compared with regular individual-level regression analyses that assume all

individuals are independent, the multilevel modeling approach accounts for the fact that people who live in the same area may be similar in some characteristics. Second, we assume that not only are the respondents in the same EA are similar in individual characteristics, but also, they share the same supply environment for FP methods. As a result, the multilevel model is appropriate to produce information on the proportion of total variation that is explained by EA-level predictors. Random-effects models typically include a random intercept and random slopes. This analysis allows for random intercepts across EAs and assumes fixed effects of covariates across EAs. The model can be shown with two equations: one at the individual level and one at the EA level.

The following equations define the fixed-effect and random-effect components of the model:

Level 1 (individual level)

$$Y_{ij} = \beta_{0j} + \beta_{1j}X_{ij} + r_{ij} \tag{1}$$

Level 2 (EA level)

$$\beta_{0j} = \gamma_{00} + \gamma_{01}Z_j + u_{0j} \tag{2}$$

$$\beta_{1j} = \gamma_{10} \tag{3}$$

Substitution of (2) and (3) in (1)

$$Y_{ij} = \gamma_{00} + \gamma_{10}X_{ij} + \gamma_{01}Z_j + r_{ij} + u_{0j}$$

Where

| | |
|---|---|
| i | : individual woman |
| j | : EA |
| $\beta_{0j}$ | : the mean of $Y_{ij}$ for EA j |
| $\gamma_{00}$ | : the grand mean of $Y_{ij}$ |
| $X_{ij}$ | : the predictor variable for the individual woman i in the EA j |
| $\gamma_{10}$ | : the effect of the predictor variable $X_{ij}$ across EAs |
| $r_{ij}$ and $u_{0j}$ | : random components normally distributed and independent of each other |

To assess the effects of EA variability on the current use of modern contraceptives among women in Kinshasa, we use Stata's multilevel analysis command *merqlogit*. Five models are fitted (presented in Box 1):

## Box 1. Fitted models and variables included in models.

| | | Models | | | | |
|---|---|---|---|---|---|---|
| | | Model 0 | Model 1 | Model 2 | Model 3 | Model 4 |
| EA-level SDP variable | | | Yes | Yes | Yes | Yes |
| Individual-level variable | | | | Yes | Yes | Yes |
| EA-level community average variables | | | | | Yes | Yes |
| Interaction of time with EA-level variables | | | | | | Yes |

**Null model:** no independent variables were included in the model. This model tests the random effect of between-cluster variability.

**Model 1:** The first model included only the EA-level FP service variables drawn from the SDP survey and measures the impact of EA-level supply and service environment on modern contraceptive use.

**Model 2:** The second model included the individual-level variables as well as EA-level FP service variables drawn from SDP to determine their combined fixed and random effects on the use of modern contraceptives.

**Model 3:** In the third model, we included the EA-level factors drawn from individual data in addition to other factors in Model 2. It thus assessed the influence of community factors on modern contraceptive use controlling for individual-level factors and EA-level service variables to determine their combined fixed and random effect on the use of modern contraceptives.

**Model 4:** This model is fitted following the primary finding that the variable representing the round of data is significantly predictive of use of modern contraceptives. This suggests that the rapidly changing supply environment may have a different influence on modern contraceptive use at different times. In the last model, we included the interaction terms between the variable for the round of data and all the EA-level variables. This model assesses the influence of community factors, EA-level service factors and individual factors alongside with the effect of time on modern contraceptive use.

A likelihood ratio test (LR test) was used to compare the goodness-of-fit of each model against the previous model. The analysis was weighted to correct for the complex survey design used by PMA2020. All statistical analyses were conducted using Stata 15.1 [38]

## Results

### A. Socio-demographic profile of female respondents and contraceptive use

Overall, three socio-demographic characteristics of women did not differ significantly across the four rounds: age, number children, and education (Table 2). Over four rounds, the distribution and average of the age of interviewed women did not change significantly (mean = 28). The majority of respondents in all four rounds had secondary or higher education, ranging from 79% to 89%. The proportion of women married or in a union in the sample was significantly different throughout the rounds ranging from 43.4% to 49.9%. The percentage of pregnant women in the sample was consistent over the four rounds (4.7–5.8%). Across four rounds of data, women's mean number of live children at the time of the interview ranged from 1.7 to 1.8. While at least 40% of women had no children, almost one third had more than three live children throughout four rounds. When women were asked if they had the desire for more children, over 78% answered positively, which was relatively consistent between rounds. Respondents were also asked if they have seen or heard anything messages about FP on television or radio or read about family planning in the newspaper in the last few months. Our analysis shows that at least one third of women had never been exposed to any FP messages. However, there was a significant change in the proportion of women who received FP

**Table 2. Individual characteristics of women PMA 2020, 2014–2016, Kinshasa, DRC.**

|  | Round 2 | Round 3 | Round 4 | Round 5 | P-value*ζ |
|---|---|---|---|---|---|
|  | % (N = 2902) | % (N = 2715) | % (N = 2756) | % (N = 2595) |  |
| **Age** |  |  |  |  | 0.603 |
| 15–24 | 42.6 | 43.3 | 41.6 | 43.0 |  |
| 25–34 | 31.4 | 29.1 | 32.5 | 30.7 |  |
| 35–49 | 26.0 | 27.6 | 25.9 | 26.3 |  |
| **Mean age** | 28.0 | 28.0 | 27.9 | 28.1 | 0.768 |
| **Education** |  |  |  |  | **<0.001** |
| Never | 1.4 | 2.3 | 3.1 | 2.2 |  |
| Primary | 9.7 | 17.5 | 21.2 | 18.6 |  |
| Secondary | 72.9 | 67.4 | 62.6 | 63.6 |  |
| Higher | 16.0 | 12.8 | 13.1 | 15.7 |  |
| **Married/in union** | 49.9 | 46.1 | 43.4 | 47.0 | **0.009** |
| **Married more than once** | 17.2 | 16.1 | 12.9 | 11.1 | 0.044 |
| **Pregnant** | 5.8 | 5.4 | 5.8 | 4.7 | 0.512 |
| **Number of live children** |  |  |  |  | 0.129 |
| 0 | 40.6 | 40.2 | 41.6 | 41.0 |  |
| 1–2 | 28.8 | 30.8 | 29.8 | 31.2 |  |
| +3 | 30.7 | 29.0 | 28.7 | 27.9 |  |
| **Mean number of children** | 1.8 | 1.7 | 1.7 | 1.7 | 0.963 |
| **Desire for more children?** |  |  |  |  | 0.492 |
| Yes | 78.1 | 79.0 | 80.7 | 79.8 |  |
| **Exposed to FP messages** | 57.2 | 56.8 | 68.0 | 67.0 | **0.001** |
| **Modern Contraceptive Use** | 16.9 | 17.0 | 20.9 | 20.9 | **0.037** |

* Chi-square test was conducted to test the significance in the difference of categorical variables across rounds of data.

ζ Analysis of variances was conducted to test the significance in the difference of continuous variables across rounds of data (age and number of children).

messages from at least one of these sources from Round 2 to Round 5. In all rounds, television was the primary source of FP messages followed by radio and magazine (results not shown). Modern contraceptive use increased significantly over the course of four rounds, with 16.9, 17.0, 20.9, and 20.9% of women of reproductive age were using a modern contraceptive method. However, modern contraceptive use among EAs varies widely from 1.5% in some EAs to 58% in others (Fig 2).

## B. Family planning service availability at Enumeration Area (EA) level

Analysis of EA-level access variables drawn from the SDP survey indicates that women in each EA have access to at least two SDPs that offer FP services. Table 3 presents the family planning service availability at the EA level. The mean number of modern methods offered in each EA ranged from 9.1 in Round 2 to 11.1 in Round 3. However, when we limit the SDPs to those that have at least five methods in stock in each EA, the average number is less than one SDP in all rounds. On average, there were 1.1 to 1.5 SDPs that offered more than three modern methods in each EA. Women's access to LAPM methods in each EA varied throughout four rounds. On average, the SDPs offered 2.1 to 2.5 LAPM methods over four rounds. Women in different EAs, on average, had access to 2.1 to 2.5 SDPs that offered LAPM. However, on average there was only one (0.8–1.1) SDP with the technical capacity necessary to insert an implant and roughly one (0.8–0.9) SDP with the technical capacity needed to insert an IUD.

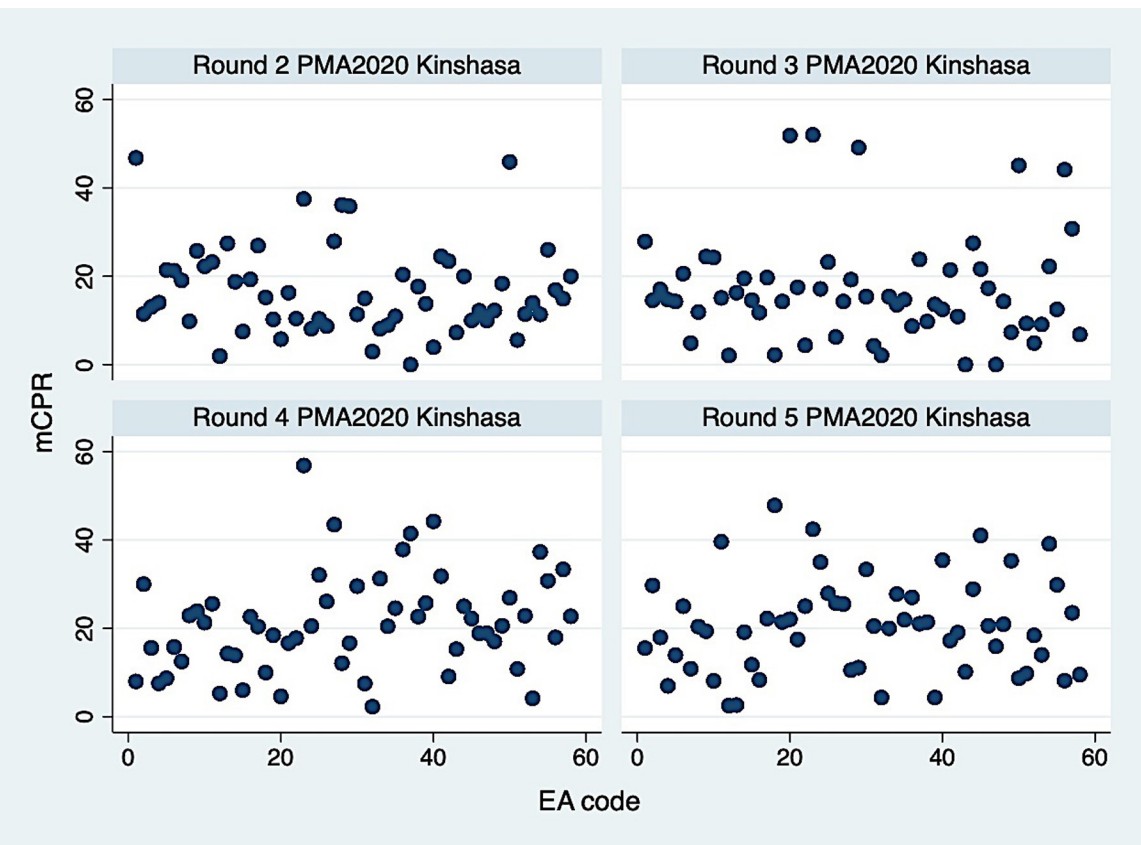

**Fig 2. Modern contraceptive prevalence by EA in round 2–5 of PMA2020 Kinshasa, DR.**

Table 4 shows results of the bivariate test of the association between modern contraceptive use and individual women's characteristics. This analysis indicates that age, marital status, and the number of live children are significantly associated with modern contraceptive use in all four rounds. The relationship between age and modern contraceptive use was curvilinear as expected, with lower use on both ends of the age spectrum.

## Multivariate multilevel analysis

We used multilevel multivariate logistic regression models to test the association between modern contraceptive use and three groups of determinants, EA-level SDP variables, community-level factors, and individual-level women's characteristics. In Model 1, we examined the association between modern contraceptive use to EA-level SDP variables. The strength of the association between the explanatory variables and contraceptive use was measured using the odds ratio (Table 5).

Multivariate analysis of modern contraceptive use and EA-level SDP variables (Model 1) identified two factors that significantly increased the odds of contraceptive use among women in Kinshasa. Women who lived in EAs with a higher number of methods offered and a higher number of SDPs with fees had higher odds of using modern contraceptives. In contrast to the bivariate analysis results, a higher number of days per week that an SDP offered FP services decreased the predicted odds of modern contraceptive use.

In Model 2, we tested the impact of EA-level SDP variables on modern contraceptive use after controlling for individual-level women's background characteristics. Three EA-level

**Table 3. Family planning service availability per EA, PMA 2020, 2014–2016, Kinshasa, DRC.**

|  | Round 2 | Round 3 | Round 4 | Round 5 |
|---|---|---|---|---|
| **EA-level SDP variables** |  |  |  |  |
| Mean number SDPs offer FP | 2.7 | 2.9 | 2.5 | 1.9 |
| Mean number method per EA | 9.1 | 11.1 | 10.3 | 9.4 |
| Mean number SDPs >3 methods | 1.1 | 1.5 | 1.3 | 1.3 |
| Mean number SDPs > 5 methods | 0.6 | 0.8 | 0.7 | 0.7 |
| Mean number days per week | 3.5 | 3.8 | 3.6 | 3.3 |
| Mean number of LAPMs* | 2.1 | 2.5 | 2.1 | 2.4 |
| Mean number of SDP insert implant | 1.0 | 1.1 | 0.9 | 1.1 |
| Mean number of SDP insert IUD | 0.8 | 0.8 | 0.8 | 0.9 |
| Mean number methods stocked-out | 1.9 | 3.3 | 3.3 | 1.7 |
| Mean number of SDP with electricity and water | 1.2 | 0.9 | 0.9 | 1.0 |
| **EA-level community variables** |  |  |  |  |
| Mean number of methods known per EA | 5.5 | 6.4 | 6.2 | 6.5 |
| Mean wealth score per EA | 0.1 | 0.0 | 0.1 | 0.2 |
| Mean age per EA | 28.0 | 28.0 | 27.9 | 28.1 |
| Mean level of education | 2.0 | 2.0 | 2.0 | 2.0 |
| Mean proportion of autonomy per EA | 0.2 | 0.2 | 0.2 | 0.2 |
| Mean parity per EA | 1.8 | 1.7 | 1.7 | 1.7 |
| *N (EA)* | 58 | 58 | 58 | 58 |

*Long-acting permanent methods.

variables and seven of individual characteristics emerged as correlates of modern contraceptive use. After controls for women's background characteristics, we found that in addition to number of SDPs with fees per EA (which was significantly associated to modern contraceptive use in Model 1), the number of SDPs that offer FP emerged as a positively associated variable to modern contraceptive use. Adding individual characteristics also yielded significant predictors of modern contraceptive use: being divorced or widowed, desire for more children, and being interviewed in Round 3 were negatively and significantly associated with modern contraceptive use. Furthermore, knowing a higher number of methods, being sexually active in the last 30 days, having a greater number of children, having secondary or advanced education, and being exposed to a higher number of FP messages through different media outlets were positive predictors of modern contraceptive use. Therefore, keeping all other variables constant, on average if a woman knows one more FP method, she has seven percentage points higher odds of using a modern contraceptive. In addition, women who have been sexually active have 34 percentage points higher odds of using modern contraceptives. Another finding in this model was that women with secondary and higher education had significantly higher odds of using modern contraceptives by approximately 40 percentage points. The analysis of marital status showed that women who have never been married have 45 percentage point higher, and women who are divorced or separated have 20 percentage points lower odds of using any modern contraceptives. Also, women who were exposed to FP messages via a higher number of media outlets (radio, television, and newspaper) showed an increased predicted odds of modern contraceptive use by more than 30%. Another finding in this model was the significant association of the time variable (round of PMA2020) in Rounds 4 and 5 with contraceptive use. This finding means that women in Round 4 and 5 had at least 15% higher odds of using modern contraceptives after controlling for individual background and EA-level SDP service factors.

**Table 4.** Association between modern contraceptive use and individual characteristics, PMA2020, Kinshasa, DRC.

| | Round 2 | P-value* | Round 3 | P-value* | Round 4 | P-value* | Round 5 | P-value* |
|---|---|---|---|---|---|---|---|---|
| **Age** | | <0.0001 | | <0.001 | | <0.001 | | 0.003 |
| 15–24 | 12.5 | | 14.3 | | 15.9 | | 16.8 | |
| 25–34 | 21.0 | | 23.5 | | 27.8 | | 26.4 | |
| 35–49 | 18.7 | | 14.4 | | 20.4 | | 21.0 | |
| **Mean age** | | <0.001 | | 0.128 | | <0.001 | | 0.044 |
| FP non-user | 27.7 | | 27.9 | | 27.6 | | 27.8 | |
| FP user | 29.6 | | 28.6 | | 29.0 | | 29.0 | |
| **Education** | | 0.210 | | 0.327 | | 0.024 | | 0.758 |
| Never | 18.0 | | 14.3 | | 31.3 | | 14.4 | |
| Primary | 19.1 | | 16.7 | | 18.0 | | 22.0 | |
| Secondary | 16.0 | | 16.2 | | 20.0 | | 20.9 | |
| Higher | 19.5 | | 21.9 | | 28.1 | | 20.8 | |
| **Married/In union** | | <0.001 | | <0.001 | | 0.021 | | 0.140 |
| Yes | 21.5 | | 20.3 | | 23.6 | | 23.4 | |
| No | 12.3 | | 14.1 | | 18.9 | | 18.7 | |
| **Mean number of Live children** | | <0.001 | | <0.001 | | <0.001 | | 0.003 |
| Non-user | 1.7 | | 1.7 | | 1.6 | | 1.6 | |
| User | 2.6 | | 2.2 | | 2.2 | | 2.1 | |
| **FP decision** | | <0.001 | | <0.001 | | <0.001 | | <0.001 |
| Individually | 39.9 | | 34.2 | | 35.8 | | 42.4 | |
| Others involved | 12.1 | | 13.0 | | 16.9 | | 15.4 | |
| **Desire for additional children?** | | <0.001 | | 0.019 | | <0.001 | | 0.001 |
| Yes | 15.1 | | 16.0 | | 19.4 | | 19.0 | |
| No | 23.3 | | 20.8 | | 27.3 | | 28.3 | |
| **Received FP ad** | | 0.002 | | .008 | | 0.011 | | 0.005 |
| Yes | 13.5 | | 13.5 | | 16.3 | | 16.4 | |
| No | 19.4 | | 19.6 | | 23.1 | | 23.1 | |
| *N* | *2902* | | *2715* | | *2756* | | *2595* | |

*Based on t-test and chi-square test. Bivariate analysis of modern contraceptive use and individual women characteristics.

In Model 3, shown in Table 5, we included EA-level community variables as well as EA-level SDP variables and individual-level characteristics of women. We found that two community-level variables were associated with modern contraceptive use. Women who lived in EAs with a higher average number of known FP methods had 21% higher odds of using modern contraceptive methods. Furthermore, living in EAs in which a higher proportion of women make the contraceptive decision was positively associated with modern contraceptive use; women in these EAs had at least six times higher odds of using modern contraceptives.

Model 4 controlled for individual variables, EA-level SDP and community variables and time interactions with EA-level SDP variables. This analysis suggested that of 11 EA-level SDP variables, two were significantly and positively associated with higher odds of modern contraceptive use. Women who resided in EAs with a higher number of LAPM methods offered had 28% higher odds of using a modern contraceptive. Similarly, women who lived in EAs with a greater number of stocked-out SDPs had 8% lower odds of using modern FP methods.

In the final model (Model 4), in which we simultaneously test the effects of three categories of variables on modern contraceptive use after controlling for the interaction of time variable (round) and the EA-level SDP variables, several EA-level SDP and community variables

**Table 5. Multivariate models for modern contraceptive use, PMA2020, Kinshasa, DRC.**

| | Model 1[η] | | Model 2[ξ] | | Model 3[ψ] | | Model 4[ς] | |
|---|---|---|---|---|---|---|---|---|
| | AOR | SE | AOR | SE | AOR | SE | AOR | SE |
| **Number of SDPs offer FP** | 1.04 | 0.05 | 1.09* | 0.06 | 1.08 | 0.06 | 1.20 | 0.18 |
| **Number of methods per EA** | 1.09** | 0.04 | 1.00 | 0.05 | 1.00 | 0.05 | 1.15 | 0.23 |
| **Number of SDPs > 3 methods** | 1.01 | 0.06 | 1.02 | 0.06 | 1.03 | 0.06 | 0.97 | 0.17 |
| **Number of SDPs > 5 methods** | 1.04 | 0.06 | 1.07 | 0.07 | 1.01 | 0.07 | 1.18 | 0.24 |
| **Number of days SDPs** | 0.92** | 0.03 | 0.94** | 0.03 | 0.95 | 0.03 | 0.86* | 0.07 |
| **Number of SDPs with LAPM** | 0.97 | 0.04 | 0.99 | 0.05 | 1.00 | 0.05 | 1.28* | 0.19 |
| **Number of SDPs insert implant** | 0.91 | 0.08 | 0.94 | 0.08 | 0.94 | 0.08 | 0.49*** | 0.11 |
| **Number of SDPs insert IUD** | 1.04 | 0.09 | 1.03 | 0.10 | 1.05 | 0.10 | 0.81 | 0.23 |
| **Number of SDPs stocked-out** | 1.01 | 0.01 | 1.00 | 0.01 | 0.99 | 0.01 | 0.92** | 0.03 |
| **Number of SDPs with fees** | 1.01** | 0.01 | 1.01** | 0.01 | 1.01 | 0.01 | 1.12 | 0.12 |
| **Number of SDPs with water and electricity** | 0.95 | 0.03 | 0.94* | 0.03 | 0.94* | 0.03 | 0.85** | 0.06 |
| **Age** | | | | | | | | |
| 15–24 | | | Ref. | | Ref. | | Ref. | |
| 25–34 | | | 1.54*** | 0.13 | 1.55*** | 0.14 | 1.57*** | 0.14 |
| 35–49 | | | 1.19 | 0.2 | 1.17 | 0.2 | 1.16 | 0.2 |
| **Level of education** | | | | | | | | |
| No education | | | Ref. | | Ref. | | Ref. | |
| Primary school | | | 0.95 | 0.17 | 0.92 | 0.17 | 0.96 | 0.18 |
| Middle secondary | | | 1.04 | 0.18 | 1.00 | 0.18 | 1.06 | 0.19 |
| Advanced secondary+ | | | 1.39* | 0.27 | 1.35 | 0.26 | 1.42 | 0.28 |
| **Marital status** | | | | | | | | |
| Married | | | Ref. | | Ref. | | Ref. | |
| Never married | | | 1.45** | 0.21 | 1.45** | 0.21 | 1.50** | 0.22 |
| Sep/div/wid | | | 0.79* | 0.11 | 0.8* | 0.11 | 0.82 | 0.11 |
| **Household wealth** | | | | | | | | |
| Quantile 1 (lowest) | | | Ref. | | Ref. | | Ref. | |
| Quantile 2 | | | 0.93 | 0.08 | 0.93 | 0.08 | 0.93 | 0.09 |
| Quantile 3 | | | 0.95 | 0.09 | 0.97 | 0.09 | 0.97 | 0.09 |
| Quantile 4 | | | 0.91 | 0.09 | 0.94 | 0.10 | 0.92 | 0.09 |
| Quantile 5 | | | 0.83* | 0.09 | 0.87 | 0.09 | 0.85 | 0.09 |
| **Number of FP methods known** | | | 1.08*** | 0.01 | 1.07*** | 0.01 | 1.07*** | 0.01 |
| **Sexually active (binary)** | | | 3.84*** | 0.25 | 3.8*** | 0.24 | 3.78*** | 0.24 |
| **Desire for more children (binary)** | | | 0.77*** | 0.06 | 0.77*** | 0.06 | 0.76*** | 0.06 |
| **Parity** | | | 1.27*** | 0.03 | 1.27*** | 0.03 | 1.27*** | 0.03 |
| **Number of FP message received** | | | | | | | | |
| 0 | | | Ref. | | Ref. | | Ref. | |
| 1 | | | 1.32*** | 0.09 | 1.31*** | 0.09 | 1.29*** | 0.09 |
| 2 | | | 1.4*** | 0.1 | 1.34*** | 0.1 | 1.33*** | 0.1 |
| 3 | | | 1.57*** | 0.19 | 1.53*** | 0.18 | 1.55*** | 0.19 |
| **Round of PMA2020** | | | | | | | | |
| 2 | | | Ref. | | Ref. | | Ref. | |
| 3 | | | 0.91 | 0.07 | 0.73*** | 0.07 | 4.5 | 13.68 |
| 4 | | | 1.15* | 0.09 | 0.93 | 0.08 | 0.07 | 0.18 |
| 5 | | | 1.18* | 0.11 | 0.95 | 0.1 | 0.73 | 1.95 |
| **EA-level number of FP methods known** | | | | | 1.21*** | 0.06 | 1 | 0.18 |
| **EA-level wealth** | | | | | 0.95 | 0.04 | 1.02 | 0.09 |

*(Continued)*

**Table 5.** (Continued)

| | Model 1[η] | | Model 2[ξ] | | Model 3[ψ] | | Model 4[ς] | |
|---|---|---|---|---|---|---|---|---|
| | AOR | SE | AOR | SE | AOR | SE | AOR | SE |
| **EA-level age** | | | | | 1.01 | 0.03 | 0.99 | 0.06 |
| **EA-level final decision on FP** | | | | | 6.06*** | 1.82 | 5.32** | 3.28 |
| **EA-level parity** | | | | | 0.92 | 0.12 | 1.05 | 0.28 |
| *N* | *10882* | | *10882* | | *10882* | | *10882* | |
| *LR chi2* | *26.77*** | | *104.42**** | | *65.92**** | | *82.33*** | |

* Significant at *p ≤ 0.10;

**p ≤0.05;

***p ≤ 0.01. **AOR**: adjusted odds ratio, SE: standard error, LAPM: long-acting permanent method.

η Controlled for EA-level SDP variables.

ξ Controlled for EA-level SDP variables and individual-level women's characteristics.

ψ Controlled for EA-level SDP variables, individual women's characteristics, and EA-level community variables.

ς Controlled EA-level SDP variables, individual women's characteristics, EA-level community variables, the interaction of time (round) and EA-level SDP variables.

emerged as significant. Individual women who reside in EAs with higher number of SDPs that provide LAPM methods had higher odds of using modern contraceptive methods. Similarly, women who lived in EAs with higher number of stocked-out SDPs had smaller odds of using modern contraceptive methods. We also found the proportion of women within an EA who make the contraceptive decision to be positively related to use.

The significance for the EA-level SDP and time variable (round) interactions in some rounds indicated that the variation in modern contraceptive use due to EA-level variables was partly related to changes in those variables over the four rounds of the survey.

## Discussion

Increasing the mCPR in a country with a fertility rate as high as that in the DRC requires a better understanding of the dynamics of modern contraceptive use and factors affecting women's contraceptive behavior. In this study, we assessed the extent to which the FP supply environment in Kinshasa affects modern contraceptive use, taking advantage of the PMA2020 surveys with four rounds of data from both women and facilities. Considering that there is a dearth of literature on community level and supply environment factors influencing contraceptive use in the DRC, the findings from this study could inform the national family planning program and other family planning stakeholders to address the supply need for contraception, community-based interventions, and the individual needs of women in order to impact their attitudes towards contraceptive use.

To determine whether EA-level SDP variables impact modern contraceptive use among women, we introduced the individual-level women's characteristics in addition to EA-level SDP variables. The variables that remained significantly associated with modern contraceptive use after controlling for women's characteristics was the number of SDPs with fees per EA (Model 2). Our analysis also indicated that among individual characteristics, having secondary or higher education, number of known FP methods, and exposure to FP messages via media were significantly associated with higher modern contraceptive use among women in reproductive age. These findings were in line with several studies that reported that education has a substantial positive impact on modern contraceptive use [39–41]. In many African countries, education is a predictor of socioeconomic status, as well as contraceptive use [42]. Therefore, women with lower educational attainments have lower uptake rates of contraceptives [43–45].

Surprisingly, our findings did not confirm the role of wealth on modern contraceptive use. This finding is in contrast with previous results of studies from other countries [14, 45, 46], but consistent with another study from the DRC [47].

We also added the EA-level community average variables to our model. The variables that were significantly associated with modern contraceptive use were the community average of knowledge of FP methods and FP final decision. this analysis identified several aspects of community context that related to modern contraceptive use: knowledge and decision-making and exposure to FP messages. Our finding showed that women who reside in EAs that had a higher proportion of women who make decisions regarding contraception have more than five times greater odds to use modern contraceptives. This finding is consistent with the results the result of Stephenson and his colleagues' study in Eastern Cape, South Africa [48]. The extent to which women make contraceptive decisions can indicate the level at which women in a society are empowered. Other studies also have found that female empowerment expands women's choices and ability to make decisions, including reproductive health decisions, and it also leads to improved health-seeking behavior such as modern contraceptive use [14]. Although many studies have investigated the role of women's empowerment in the decision-making process [49], they assessed other aspects of the decision-making such as the decision to seek health care, household decisions, financial decisions in addition to reproductive decision [50–52]. Our analysis found a positive association between community-level contraceptive decision and modern contraceptive use; however, we did not have the means to further investigate the women's empowerment more deeply.

Likewise, regarding the knowledge of contraceptives at the community level and exposure to FP messages, our findings are consistent with other previous results that found that exposure to mass media has substantial effects on attitudes towards family planning use among women [53–56]. Also, another previous finding suggests that modern contraceptive use is higher when the demand is generated for FP, and women are exposed to the FP messages [57]. Our finding further supports the conventional idea that women's use of modern contraceptive methods increases with the increase in parity [58–60]. Also, we found that sexually active and single women have higher odds of using modern contraceptives compared to married women. This finding corroborates results from other studies that found that sexually active single women in Africa have greater likelihood of using contraceptives compared to married women [61, 62].

We finally introduced the interaction of the time variable (round of data) with the EA-level SDP variables as well as EA-level community average variable to the model. Inclusion of the interaction terms does not dramatically change our results. However, the final model showed that after including all three groups of variables, the mean number of SDPs with LAPM methods were significantly associated with modern contraceptive use. Also, women in EAs with higher mean number stocked-out SDPs have lower odds of modern contraceptive use. This finding indicates that among the different access measures we applied in the study, two indicators of service availability (availability of LAPM methods and having the methods in stock) influence modern contraceptive use among women. These findings align with results from Wang and colleagues who used the DHS and SPA from Kenya, Tanzania, Uganda, and Rwanda to examine the extent to which contraceptive use is associated with the regional family planning supply and service environment [14].

This analysis has multiple limitations and assumptions which stem from our data and methodology. First, this analysis is based on the assumption that official boundaries for EAs are the same as unofficial communities. Using enumeration areas as a proxy for communities is a common practice. However, we know that the enumeration areas do not equate to the actual communities. These administrative boundaries do not fully capture the socio-cultural

characteristics of the population of their residents [63]. Our analysis used the predetermined EAs as an approximation to communities to link the availability of FP services to service utilization, however, we are limited because these official units do not account for social interaction criteria used in defining a neighborhood or community [64].

Second, in this analysis, we assumed that women in each EA would utilize the FP services from the same EA. Therefore, we expect any changes in supply environment in an EA to impact the modern contraceptive use in that EA. However, both qualitative and quantitative studies in the DRC and other sub-Saharan African countries have shown that women prefer to bypass the closest facility to acquire their desired method from a farther facility [65–68]. This can be the product of a lack of confidence in the availability and quality of service in the closest facility [68, 69]. Also, sociocultural norms can be a powerful driver for women to bypass the closest facility to avoid encountering family and friends.

Third, the information gathered through the SDP survey is cross-sectional data and not a full picture of the ever-changing supply environment in Kinshasa. Most of the supply chain is managed by multiple donors and family planning implementing organizations which procure and distribute the commodities through parallel channels in the national health system [70]. In addition to this, the structure of service delivery in the DRC (similar to most LMICs) consists of fixed facilities, pharmacies, community-based distribution workers, unofficial drug shops, and campaign days. The SDP survey does not capture information related to the activities of organizations on campaign days, community-based distribution, and most of the unofficial drug shops.

Our findings further indicate that the applied elements of access using PMA2020 as the source of data is not necessarily sufficient to monitor FP2020 goal achievement. The FP2020 monitoring framework consists of a set of a indicators captured by some of these six elements (for example, contraceptive supply stock-out and contraceptive supply availability). Many other sources of data (service statistics, client exit interview, or administrative information) would be required to accurately capture all elements of access. Despite our effort to include all six elements of access (cognitive, psychological, economic, spatial, administrative, and service quality) from both population and SDP surveys, our variables do not completely capture women's access in Kinshasa.

## Conclusion

Using population- and facility-based data, we were able to account for the effect of the supply environment on modern contraceptive use. Having a higher number of SDPs with LAPM methods and fewer SDPs with stockouts were determinants of contraceptive use among women in Kinshasa. However, several SDP variables were not significantly associated with contraceptive use, and having more accurate information to measure access could be beneficial in determining the factors that impact modern contraceptive use.

In terms of the policy and programming implications, our findings suggest that availability of FP services with a broad array of method mix will increase the odds of modern contraceptive use among women in Kinshasa. Our results showed that having methods that are more desired among women (LAPM methods) increase women's contraceptive use, whereas having stockout decreases their odds to use modern contraceptives. This finding emphasizes the importance of reliable availability of methods as well as the availability of the methods that women prefer.

The EA-level effects on contraceptive behavior suggest a need for family planning programming to shift focus to community-oriented practices and service delivery (for example, FP campaigns with the focus on social and behavioral change programs). Our findings also

suggest that if women are involved in the contraceptive decision, they have greater odds of using modern contraceptives. We suggest that future studies include more questions regarding women's roles in household decision-making as well as personal and health-related decisions.

Finally, the lack of significance of most of our EA-level constructed variables suggest that we lack knowledge on both sides of the supply and demand equation. We recommend that international FP stakeholders reach a consensus on the elements of access and measurement of these elements.

## Acknowledgments

Dr. Patrick Kayembe passed away before the submission of the final version of this manuscript. Saleh Babazadeh accepts responsibility for the integrity and validity of the data collected and analyzed. The authors wish to acknowledge the pioneering role that Dr. Kayembe played in advancing reproductive health research in the DRC.

## Author Contributions

**Conceptualization:** Saleh Babazadeh, Jane T. Bertrand.

**Data curation:** Philip Anglewicz, Patrick K. Kayembe.

**Formal analysis:** Saleh Babazadeh.

**Methodology:** Philip Anglewicz.

**Supervision:** Jane T. Bertrand.

**Writing – original draft:** Saleh Babazadeh.

**Writing – review & editing:** Saleh Babazadeh, Philip Anglewicz, Janna M. Wisniewski, Julie Hernandez, Jane T. Bertrand.

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
