## [Decision Letter · Decision Letter 0]

4 Nov 2019

PONE-D-19-26324

The Influence of Health Facility-Level Access Measures on Modern Contraceptive Use in Kinshasa, DRC.

PLOS ONE

Dear Dr. Babazadeh,

Thank you for submitting your manuscript to PLOS ONE. After careful consideration, we feel that it has merit but does not fully meet PLOS ONE’s publication criteria as it currently stands. Therefore, we invite you to submit a revised version of the manuscript that addresses the points raised during the review process.

The manuscript was reviewed by two reviewers and their comments are appended below. Both reviewers have raised number of issues. I am sure that the reviewers comments would be very helpful  to revise your manuscript. 

We would appreciate receiving your revised manuscript by Dec 19 2019 11:59PM. To enhance the reproducibility of your results, we recommend that if applicable you deposit your laboratory protocols in protocols.io, where a protocol can be assigned its own identifier (DOI) such that it can be cited independently in the future. For instructions see: http://journals.plos.org/plosone/s/submission-guidelines#loc-laboratory-protocols

We look forward to receiving your revised manuscript.

Kind regards,

Kannan Navaneetham

Academic Editor

PLOS ONE

Journal Requirements:

2. In ethics statement in the manuscript and in the online submission form, please provide additional information about the database used in your retrospective study. Specifically, please ensure that you have discussed whether all data were fully anonymized before you accessed them and/or whether the IRB or ethics committee waived the requirement for informed consent. If patients provided informed written consent to have their data used in research, please include this information.

3. Please remove your figures from within your manuscript file, leaving only the individual TIFF/EPS image files, uploaded separately.  These will be automatically included in the reviewers’ PDF.

4. Please ensure that you include a title page within your main document. You should list all authors and all affiliations as per our author instructions and clearly indicate the corresponding author.

6. We note that you have indicated that data from this study are available upon request. PLOS only allows data to be available upon request if there are legal or ethical restrictions on sharing data publicly. For information on unacceptable data access restrictions, please see http://journals.plos.org/plosone/s/data-availability#loc-unacceptable-data-access-restrictions.

Reviewers' comments:

Reviewer's Responses to Questions

**Comments to the Author**

1. Is the manuscript technically sound, and do the data support the conclusions?

Reviewer #1: Partly

Reviewer #2: Yes

2. Has the statistical analysis been performed appropriately and rigorously? 

Reviewer #1: No

Reviewer #2: Yes

3. Have the authors made all data underlying the findings in their manuscript fully available?

Reviewer #1: Yes

Reviewer #2: No

4. Is the manuscript presented in an intelligible fashion and written in standard English?

Reviewer #1: Yes

Reviewer #2: Yes

5. Review Comments to the Author

Reviewer #1: The paper linked PMA household data and health facility data to assess the association between women’s contraceptive use and different types of access measured at the EA level, as well as other community-level factors aggregated from individual data. The findings could have important implications for family planning programs in Kinshasa, DRC.

Hope the following comments could help strengthen the paper:

Background:

a) Line #51: the statement “the spatial data from the SPA and DHS have been randomly displaced” is not correct. Geolocations of DHS clusters are displaced but not those of health facilities in SPA surveys. The DHS Program released GPS data collected in SPA without displacement.

b) Line #56: I would not say “the SPA and DHS sampling strategy make it impossible to link at the EA level”. Actually, it is possible to link them at the EA level when the SPA survey is a facility census. Please refer to this report for more detail (https://www.dhsprogram.com/publications/publication-SAR10-Spatial-Analysis-Reports.cfm)

c) Line #58: I don’t think this recommendation was made in the reference #9. Please double check.

d) Study setting: it would be helpful to include some information on contraceptive method mix in Kinshasa. Do more women use LARCs? This may help interpret the findings on the association that having more facilities with LARCs increases the likelihood of contraceptive use.

Methods:

Conceptual framework: this paragraph reads more like a description of different types of access (i.e. their definitions) rather than a conceptual framework. A conceptual framework would demonstrate/discuss the pathway by which variables, alone or in combination, are associated with the outcome of interest. Also it would be helpful to map the SDP-level variables and other community-level variables introduced later to these different “access” categories.

Data:

a) The sample design of the 4 rounds was independent? Or the same EAs were followed over the 4 rounds? This is not clear.

b) It would be helpful to provide some information on the total number of SDPs in EAs from which the SDP sample were selected. Is it possible that same facilities, such as tertiary hospitals are shared by multiple EAs? If this is the case, how did you deal with this in the analysis?

Key independent variables:

a) SDP variables---I have concerns about the SDP variables constructed at the EA level by counting the total number of SDPs. These include the total number of SDPs that off FP methods, total number of SDPs with more than 3 methods, total number of SDPs with more than 5 methods in stock, and several others. These SDPs were a sample of all SDPs that serve the EA. The total count may not represent the service environment unless you count among all SDPs that serve the EA. Plus you have a max value of 6 for all these indicators, right? Because the sample size of SDPs would not exceed 6 in each EA. Instead of counting, for example the total number of SDPs that offer at least 3 FP methods, a proportion measurement, i.e. proportion of SDPs with at least 3 methods might be a better measurement.

Moreover, in the analysis, do you treat these variables as continuous variable? I guess their possible values would be 0, 1 ,2 ,3 ,4 ,5 ,6? They are not continuous.

b) Community-level variables constructed with individual data --- the decision-making variable is not clear. Decision making for what?

Analysis

a) I assumed you pooled data from 4 rounds for the multivariate analysis. Please make this clear in the description

b) I am concerned about putting all the SDP variables in the same model. Did you check the correlation among them? Some may be highly correlated for example, number of facilities with 3 methods and number of facilities with 5 methods. Others could be highly correlated too. I would double check. I wonder if this is the reason for the lack of significance of many SDP variables.

c) Line #239. It should be noted as formula (3)

d) Line #241: It should not be noted as formula (3)

e) Was the complex survey design accounted for in the analysis? The paper did not mention this.

Results

a) Table 2: what is LAPM? This term appears in several places. Please correct/clarify.

b) Line #343-353: where are the results described in this paragraph?

c) Table 4: I did not see the results on the interaction terms. The authors said something about such results in Line #435. But it is not clear which EA-level variables affect contraceptive use differently in the four rounds.

d) Line 387: the association between contraceptive use and number of facilities with water and electricity was negative not positive (OR<1).

e) The authors often use “x% more likely” in the interpretation of odds ratios. This is not correct. These are odds ratios not probabilities. It should be interpreted as something like this: the odds of using contraception is x% more/higher for group A compared to Group B.

f) I did not see where in the results showed the answers to Research Question 2) and 3). To answer these questions, the authors need to assess the additional variation explained by adding another group of variables, for example, from model 2 to model 3, how much (e.g. %) additional variation is explained by adding other community-level variables.

Discussion

a) Line 473: some references seem to be missing here.

b) The authors may want to discuss a few surprising findings from the multivariate analysis: these include: the negative association between contraceptive use and number of facilities with water and electricity in models 2, 3, and 4, negative association between contraceptive use and number of SDPs that insert implant in model 4.

Reviewer #2: I would suggest that the authors frame the introduction and the paper in the context of the Democratic Republic of Congo (DRC) as opposed to what the world as a whole is struggling with. The text in the paragraph on the study setting (i.e., Line 89 - Line 106) could be used to introduce the problem because it is specific to DRC. The authors could then introduce the influence of the Sustainable Development Goals (SDGs) as they have bearing on the way a country like the DRC shapes its health policy. Then the authors could launch into a discussion of how challenging it is to measure or conceptualize access as they did in what is now the third paragraph of the introduction (i.e., Line 42 - Line 58). That lays the groundwork for a statement of objectives and manuscript aims.

The authors state their study aims as bullets (Line 69 - Line 75). I would recommend they weave those statements into the narrative as opposed to stating them separately.

I recommend the authors define their acronyms throughout the paper. Their manuscript makes use of several acronyms and it can make for confusing reading when the meaning of certain acronyms is not clear. For example, Line 56 they methionine EA-Level long before they define it as Enumeration Area in the method section.

The authors should not forget to provide units for measures of certain demographic characteristics. For example, they state the fertility rate as 6.3 (Line 91). Perhaps authors could add births per woman as the unit the first time they report TFR.

Outlining a conceptual framework is most helpful. However, reading through what the authors set out in the methods, I am uncertain as to whether it is a single comprehensive framework or if the authors are cobbling together several constructs from different studies (I am not deeply familiar with Choi et al., 2016). If it is a single comprehensive framework, then the authors should consider describing how the various access constructs are related to each other, or not. And if these are constructs coming together from several studies, then it may be useful for the authors to characterize this properly. For example, in one study from a specific setting, access to mCPR was characterized as Administrative, while in other contexts it is characterized as Psychological access, and so on.

In Line 154, the authors describe the different types of service delivery points (SDPs). They conclude the list by writing, “and others”. It is a bit of a vague characterization. I recommend the authors revisit it.

The methods section the authors describe the explanatory variables, grouping them as availability of mCPRs, contextual factors, and individual-level factors. Within these three groups, I would suggest that they also highlight where these variables fit in the context of the framework, as they summarize in Table A1 of the appendix. By the way, it may be helpful to include that table in the manuscript as opposed to relegating it to the appendix. The table is a nice summary of the variables and is a lot clearer than the textual description provided in the manuscript.

I suggest that the authors move the content of Lines 226 - 228, to the results section. They are a description of the setting derived from analysis of the data similar to their report of individual-level characteristics or the levels of family planning service availability. In reporting the prevalence of contraceptive use by enumeration area (EA), I recommend the authors report the mean prevalence rate and perhaps the range, for each of the rounds of PMA2020. As opposed to just stating that rates ranged from 8% in some to 58% in others.

In Line 275, the authors should consider providing a full citation of STATA (and make sure it is reflected in the bibliography).

For Table 3, I would suggest that the authors show p-values as a footnote like they did for Table 4, as opposed to noting the specified value in the table.

In Table 3, two variables are listed under Mean Age as Non-user and User. Does this mean non-user of modern contraceptives and users of modern contraceptives? It is a bit unclear from the table.

None of the results make mention of household wealth index or mean wealth index at each EA, but the authors mention them as explanatory variables in the methods section (i.e., Line 199 and Line 194 respectively). If those variables are excluded from the analysis, then the authors may want to explain why or leave them out of the methods section altogether.

In the results section, under multivariate multilevel analysis, the authors repeatedly describe their findings as correlations (i.e., the correlation between mCPR and the explanatory variables). Regression is not correlation and I believe the appropriate term used is "association”. The authors may want to check their language here.

I would suggest that the authors shorten their results sections. Much of the points of interest described in text form are summarized neatly in the Tables, particularly Tables 3 and 4. Rather than report on each statistically significant finding, I suggest the authors pick one or two they wish to highlight and use those to frame the arguments they make in the introduction, and those points that are worth revisiting in the discussion.

An issue worth some discussion is whether the forms of access as defined in the study can be ranked (or even whether they should be). For example, does psychological access carry greater weight than service quality of cognitive ability? A ranking of these constructs within the framework of access may offer some insight into additional lines of investigation when considering this issue of access.

I think the discussion section could benefit from an examination of the results mean for the FP2020 Initiative. The authors make note of which associations were statistically significant. But what is the import of such findings? What is the practical significance of demonstrating that in the DRC, mCPR use increases with parity? What does such a finding mean for what can be done to help the country achieve its goal of lowering the fertility rate?

6. PLOS authors have the option to publish the peer review history of their article (what does this mean?). If published, this will include your full peer review and any attached files.

Reviewer #1: No

Reviewer #2: No

---

## [Author Response · Author response to Decision Letter 0]

16 Apr 2020

We want to thank the editor and both of the reviewers for reviewing our manuscript. Your review adds a tremendous value to our manuscript. The following is our response to the reviewers:

Reviewer #1: 

Reviewer #1: The paper linked PMA household data and health facility data to assess the association between women's contraceptive use and different types of access measured at the EA level, as well as other community-level factors aggregated from individual data. The findings could have important implications for family planning programs in Kinshasa, DRC.

Hope the following comments could help strengthen the paper:

Background:

a) Line #51: the statement "the spatial data from the SPA and DHS have been randomly displaced" is not correct. Geolocations of DHS clusters are displaced but not those of health facilities in SPA surveys. The DHS Program released GPS data collected in SPA without displacement.

- The sentence is revised to: "One drawback is the fact that the spatial data from the DHS have been randomly displaced as a confidentiality measure and few SPA surveys have a GPS coordinate data component" 

b) Line #56: I would not say "the SPA and DHS sampling strategy make it impossible to link at the EA level". Actually, it is possible to link them at the EA level when the SPA survey is a facility census. Please refer to this report for more detail (https://www.dhsprogram.com/publications/publication-SAR10-Spatial-Analysis-Reports.cfm)

- We changed the sentence to be more accurate. The census sampling may reduce the bias, however not all SPA surveys follow the census sampling strategy. 

- The sentences read: "The SPA and DHS sampling strategy make it difficult to link population and facilities at the EA level. To be able to make this happen, the SPA would need to be conducted in the same EAs as the DHS survey and a census sampling of facilities."

c) Line #58: I don't think this recommendation was made in the reference #9. Please double check.

- Wang and colleagues have mentioned this under the section 1.3: "Constraints in Linking Data from Facilities Surveys and Data from Population-Based Surveys"

d) Study setting: it would be helpful to include some information on contraceptive method mix in Kinshasa. Do more women use LARCs? This may help interpret the findings on the association that having more facilities with LARCs increases the likelihood of contraceptive use.

We included the following sentence and related citation: "The method mix among women using modern contraceptives in Kinshasa shows that 55% used Long-Acting Reversible Contraceptive (LARC) methods (injectables, implant, IUD), 43% used other short-acting modern methods, and less than two percent used male or female sterilization [31]"

https://www.pma2020.org/sites/default/files/PMA2018-Kinshasa-R7-FP-Brief-20190201_EN.pdf

Methods:

Conceptual framework: this paragraph reads more like a description of different types of access (i.e. their definitions) rather than a conceptual framework. A conceptual framework would demonstrate/discuss the pathway by which variables, alone or in combination, are associated with the outcome of interest. Also it would be helpful to map the SDP-level variables and other community-level variables introduced later to these different "access" categories.

We decided to add the previously omitted diagram of our conceptual framework based on your recommendation. We believe that the framework would contribute to the method section if accompanied with the table of variables and their relative access measure. Figure 1 shows the conceptual framework of the current study.

Data:

a) The sample design of the 4 rounds was independent? Or the same EAs were followed over the 4 rounds? This is not clear.

The sampling design has been consistent throughout 4 rounds. We have clarified the this fact in the following added sentence:

". The surveys were conducted in the same 58 enumeration areas (EA) in four annual and semi-annual rounds (rounds 2-5) between 2014 and 2016 (rounds three and four were conducted in 2015). Therefore, the sample is consisting of a panel of 58 EA selected in four rounds of data collection."

b) It would be helpful to provide some information on the total number of SDPs in EAs from which the SDP sample were selected. Is it possible that same facilities, such as tertiary hospitals are shared by multiple EAs? If this is the case, how did you deal with this in the analysis?

The SDPs were selected from a list of SDPs that was provided by the DRC Ministry of Health. Each SDP type was selected in a selected EA if it served that EA. For example, there is only one tertiary hospital in Kinshasa. Thus, this hospital has been selected in all rounds of the data collection as one SDP that serves all EA. Consecuently, where there is always three selected private SDPs in each EA, in some EAs we have selevted less than three public SDPs. We have added the following sentences to clarify:

 "Following this sampling approach, some SDPs have been selected in our sample more than once. However, since they are a small fraction of all SDPs (less than 8% of all SDPs), and deleting them from the analysis did not make any significant changes, we decided to keep the SDPs that where selected in multiple rounds. "

Key independent variables:

a) SDP variables---I have concerns about the SDP variables constructed at the EA level by counting the total number of SDPs. These include the total number of SDPs that off FP methods, total number of SDPs with more than 3 methods, total number of SDPs with more than 5 methods in stock, and several others. These SDPs were a sample of all SDPs that serve the EA. The total count may not represent the service environment unless you count among all SDPs that serve the EA. Plus you have a max value of 6 for all these indicators, right? Because the sample size of SDPs would not exceed 6 in each EA. Instead of counting, for example the total number of SDPs that offer at least 3 FP methods, a proportion measurement, i.e. proportion of SDPs with at least 3 methods might be a better measurement.

Moreover, in the analysis, do you treat these variables as continuous variable? I guess their possible values would be 0, 1 ,2 ,3 ,4 ,5 ,6? They are not continuous.

We appreciate your valid points about the constructed EA variable. However, due to some constraints in the sampling method (not knowing the real denominator for the SDPs and few number of SDPs with certain charecteristics) and for practical reasons, we decided to use the total number versus the proportion of the SDPs for some of the SDP characteristics. For three of the SDP variables (number of days of service, number of methods provided, and number of methods stocked out) we used the average values for all SDPs within an enumeration area. Furthermore, since the number of SDPs within each EA is a limited small number (here is 6), the percentage of SDPs produce a less accuaret measure and 

b) Community-level variables constructed with individual data --- the decision-making variable is not clear. Decision making for what?

This variable is constructed from the question "Who made the final decision for using modern contraceptive method on current or previous FP use?". We have relocated the table of variables from the appendix to the methods section to clarify the variables used in the analysis.

Analysis

a) I assumed you pooled data from 4 rounds for the multivariate analysis. Please make this clear in the description.

We clarified this fact. The first sentence in the analysis section now reads:" Descriptive analysis is performed for both individual and EA-level variables on a pooled cross-section data from the four rounds of PMA2020 in Kinshasa. We perform the analysis with tabulation of background characteristics for our study population in each round of data."

b) I am concerned about putting all the SDP variables in the same model. Did you check the correlation among them? Some may be highly correlated for example, number of facilities with 3 methods and number of facilities with 5 methods. Others could be highly correlated too. I would double check. I wonder if this is the reason for the lack of significance of many SDP variables.

We appreciate your valid point. However, only three of the SDP-variables were slightly correlated that excluding them did not change any significant results. Therefore we decided to keep them in the model since most of them are among important monitoring indicators.

c) Line #239. It should be noted as formula (3)

Thanks for Catching this error. It is corrected now. 

(1) Yij = β0j + β1jXij + rij 

Level 2 (EA level)

(2) β0j = γ00 + γ01Zj + u0j 

 (3) β1j = γ10 

Substitution of (2 ) and ( 3 ) in ( 1 )

Yij = γ00 + γ10Xij + γ01Zj + rij + u0j

d) Line #241: It should not be noted as formula (3)

Corrected. See above (c). 

e) Was the complex survey design accounted for in the analysis? The paper did not mention this.

We added the following sentence: "The analysis was weighted to correct for the complex survey design used by PMA2020. "

Results:

a) Table 2: what is LAPM? This term appears in several places. Please correct/clarify.

The footnote is added to the table to clarify. 

b) Line #343-353: where are the results described in this paragraph?

 This paragraph is describing the results from the bivariate analysis of modern contraceptive use and SDP variables. However, we have decided to omit the table from the final draft to save some space. Since none of the SDP variables were significantly associated with modern contraceptive use in all rounds, we are going to delete this paragraph.

c) Table 4: I did not see the results on the interaction terms. The authors said something about such results in Line #435. But it is not clear which EA-level variables affect contraceptive use differently in the four rounds.

The interaction of the time variable (round) and community level variables were used to control the changes in those variables over time. We have not shown the coefficients for the interaction terms to save some space in the table. We added a note for this problem to clarify. 

d) Line 387: the association between contraceptive use and number of facilities with water and electricity was negative not positive (OR<1).

We deleted the sentence. 

e) The authors often use "x% more likely" in the interpretation of odds ratios. This is not correct. These are odds ratios not probabilities. It should be interpreted as something like this: the odds of using contraception is x% more/higher for group A compared to Group B.

We corrected this issue throughout the text. 

f) I did not see where in the results showed the answers to Research Question 2) and 3). To answer these questions, the authors need to assess the additional variation explained by adding another group of variables, for example, from model 2 to model 3, how much (e.g. %) additional variation is explained by adding other community-level variables.

We have showed this fact with reporting the likelihood ratio test (LR) to statistically test the goodness of fit of each model against the previous. We also added the following sentences in the text to clarify the results. However, in logistic regression (contrary to OLS regression), creating a statistic that provides the same information as R2 is difficult. Our result shows that in each model, we have a significant likelihood ratio value which means that adding the new set of variables would still benefit the goodness of fit in our model. 

Discussion

a) Line 473: some references seem to be missing here.

We added the reference.

Reviewer #2:

1. I would suggest that the authors frame the introduction and the paper in the context of the Democratic Republic of Congo (DRC) as opposed to what the world as a whole is struggling with. The text in the paragraph on the study setting (i.e., Line 89 - Line 106) could be used to introduce the problem because it is specific to DRC. The authors could then introduce the influence of the Sustainable Development Goals (SDGs) as they have bearing on the way a country like the DRC shapes its health policy. Then the authors could launch into a discussion of how challenging it is to measure or conceptualize access as they did in what is now the third paragraph of the introduction (i.e., Line 42 - Line 58). That lays the groundwork for a statement of objectives and manuscript aims. 

We appreciate your recommendation. We believe this would help the reader to better understand the aim of this study. We have changed the introduction accordingly. We have integrated the "Stduy setting" section in the "Background" according to your recommended outline.

2. The authors state their study aims as bullets (Line 69 - Line 75). I would recommend they weave those statements into the narrative as opposed to stating them separately.

We followed the recommendation. The new pragarph reads: "In this study we aimed to investigate the extent to which EA-level FP supply and services impact the modern contraceptive use among women in reproductive age in Kinshasa, DRC. Also, we aimed to assess if variability in contraceptive use among women in reproductive age in Kinshasa can be explained by differences in EA-level contextual factor variables."

3. I recommend the authors define their acronyms throughout the paper. Their manuscript makes use of several acronyms and it can make for confusing reading when the meaning of certain acronyms is not clear. For example, Line 56 they methionine EA-Level long before they define it as Enumeration Area in the method section.

We appreciate you catching this issue. We tried to spell out each acronym the first time it shows up in the text. We checked the text again to follow our rule.

4. The authors should not forget to provide units for measures of certain demographic characteristics. For example, they state the fertility rate as 6.3 (Line 91). Perhaps authors could add births per woman as the unit the first time they report TFR.

We added units of measures where it was necessary per your recommendation.

5. Outlining a conceptual framework is most helpful. However, reading through what the authors set out in the methods, I am uncertain as to whether it is a single comprehensive framework or if the authors are cobbling together several constructs from different studies (I am not deeply familiar with Choi et al., 2016). If it is a single comprehensive framework, then the authors should consider describing how the various access constructs are related to each other, or not. And if these are constructs coming together from several studies, then it may be useful for the authors to characterize this properly. For example, in one study from a specific setting, access to mCPR was characterized as Administrative, while in other contexts it is characterized as Psychological access, and so on.

The conceptual framework was adopted from the Choi et al. study. Choi et al. synthesized the key access elements for measurement by reviewing three well‐known frameworks. All the elements used in their study were common among the three frameworks (Penchansky and Thomas 1981, Bertrand et al. 1995, and AAAQ proposed by United Nations Committee on Economic, Social, and Cultural Rights in 2000). Choi et al. used the DHS survey to measure access based on their synthesized framework. 

We used their synthesized framework to measure the elements of the access using PMA2020 population and facility surveys. Choi et al have discussed the limitations if measuring the elements of access based on solely on source of data (the DHS survey). Similarly, we have mentioned the limitations to our study, although, we used two sources of data (population and facility surveys). 

In order to clarify the conceptual framework, we decided to add the previously omitted diagram of our conceptual framework based on your recommendation. We believe that the framework would contribute to the method section if accompanied with the table of variables and their relative access measure. 

6. In Line 154, the authors describe the different types of service delivery points (SDPs). They conclude the list by writing, "and others". It is a bit of a vague characterization. I recommend the authors revisit it.

We clarified the sentence by providing an example. The new sentence reads: "Private SDPs included faith-based SDPs, pharmacies, clinics, and other unofficial providers of FP methods (such as kiosks)."

7. The methods section the authors describe the explanatory variables, grouping them as availability of mCPRs, contextual factors, and individual-level factors. Within these three groups, I would suggest that they also highlight where these variables fit in the context of the framework, as they summarize in Table A1 of the appendix. By the way, it may be helpful to include that table in the manuscript as opposed to relegating it to the appendix. The table is a nice summary of the variables and is a lot clearer than the textual description provided in the manuscript.

 We appreciate you recommendation. Per your advice, we included the appendix table within the manuscript to clarify the relation of the variable used in the analysis to the conceptual framework. 

8. I suggest that the authors move the content of Lines 226 - 228, to the results section. They are a description of the setting derived from analysis of the data similar to their report of individual-level characteristics or the levels of family planning service availability. In reporting the prevalence of contraceptive use by enumeration area (EA), I recommend the authors report the mean prevalence rate and perhaps the range, for each of the rounds of PMA2020. As opposed to just stating that rates ranged from 8% in some to 58% in others.

9. In Line 275, the authors should consider providing a full citation of STATA (and make sure it is reflected in the bibliography).

We added the proper citation.

10. For Table 3, I would suggest that the authors show p-values as a footnote like they did for Table 4, as opposed to noting the specified value in the table.

In Table 3, two variables are listed under Mean Age as Non-user and User. Does this mean non-user of modern contraceptives and users of modern contraceptives? It is a bit unclear from the table.

We changed the categories to "FP user" and "FP nonuser" to be more clear.

11. None of the results make mention of household wealth index or mean wealth index at each EA, but the authors mention them as explanatory variables in the methods section (i.e., Line 199 and Line 194 respectively). If those variables are excluded from the analysis, then the authors may want to explain why or leave them out of the methods section altogether.

We have variables of household wealth at individual level and community level in our final fitted models. However, the wealth variables have no association to modern contraceptive use. In the second paragraph of the discussion section we have stated: "Surprisingly, our findings did not confirm the role of wealth on modern contraceptive use. This finding is in contrast with previous results of studies from other countries [9, 47, 48], but consistent with another study from the DRC[49]."

12. In the results section, under multivariate multilevel analysis, the authors repeatedly describe their findings as correlations (i.e., the correlation between mCPR and the explanatory variables). Regression is not correlation and I believe the appropriate term used is "association". The authors may want to check their language here.

We appreciate your correction. We changed the language throughout the manuscript. 

13. I would suggest that the authors shorten their results sections. Much of the points of interest described in text form are summarized neatly in the Tables, particularly Tables 3 and 4. Rather than report on each statistically significant finding, I suggest the authors pick one or two they wish to highlight and use those to frame the arguments they make in the introduction, and those points that are worth revisiting in the discussion.

We appreciate your comment. Per your recommendation we have shortened the results section to cover the most important findings. 

14. An issue worth some discussion is whether the forms of access as defined in the study can be ranked (or even whether they should be). For example, does psychological access carry greater weight than service quality of cognitive ability? A ranking of these constructs within the framework of access may offer some insight into additional lines of investigation when considering this issue of access.

Authors appreciate the reviewer mentioning this point. The present study attempted to shed light on the relationship of some of the supply environment access measures rather than provide a conceptual framework. We agree with the reviewer's comment that the ranking of these access constructs is worth investigating. However, this research question is beyond the scope of this study. 

15. I think the discussion section could benefit from an examination of the results mean for the FP2020 Initiative. The authors make note of which associations were statistically significant. But what is the import of such findings? What is the practical significance of demonstrating that in the DRC, mCPR use increases with parity? What does such a finding mean for what can be done to help the country achieve its goal of lowering the fertility rate?

We added the following to clarify the discussion we had previously. We added: "Our findings further indicate that the applied elements of access using PMA2020 as the source of data is not necessarily sufficient to monitor FP2020 goal achievement. Whereas, FP2020 monitoring framework consist of a set of a indicators captured by some of these six elements (For example, contraceptive supply stock-out and contraceptive supply availability). Many other sources of data (service statistics, client exit interview, or administrative information) are required to capture all elements of access accurately."

---

## [Decision Letter · Decision Letter 1]

6 May 2020

PONE-D-19-26324R1

The Influence of Health Facility-Level Access Measures on Modern Contraceptive Use in Kinshasa, DRC.

PLOS ONE

Dear Dr. Babazadeh,

Thank you for submitting your manuscript to PLOS ONE. After careful consideration, we feel that it has merit but does not fully meet PLOS ONE’s publication criteria as it currently stands. Therefore, we invite you to submit a revised version of the manuscript that addresses the points raised during the review process.

The manuscript is sent to the reviewer who reviewed the previous version. The reviewer has suggested some minor issues which needs to be addressed. The reviewer comments is appended below. Moreover, the manuscript needs to be edited by a professional editor. In some places it is not readable that could be improved for the benefit of readers, though the meaning is clear for the authors.  I am sure that the comments by the reviewer would help to improve the quality of the manuscript.

We would appreciate receiving your revised manuscript by Jun 20 2020 11:59PM. To enhance the reproducibility of your results, we recommend that if applicable you deposit your laboratory protocols in protocols.io, where a protocol can be assigned its own identifier (DOI) such that it can be cited independently in the future. For instructions see: http://journals.plos.org/plosone/s/submission-guidelines#loc-laboratory-protocols

We look forward to receiving your revised manuscript.

Kind regards,

Kannan Navaneetham, PhD

Academic Editor

PLOS ONE

Reviewers' comments:

Reviewer's Responses to Questions

**Comments to the Author**

1. If the authors have adequately addressed your comments raised in a previous round of review and you feel that this manuscript is now acceptable for publication, you may indicate that here to bypass the “Comments to the Author” section, enter your conflict of interest statement in the “Confidential to Editor” section, and submit your "Accept" recommendation.

Reviewer #2: All comments have been addressed

2. Is the manuscript technically sound, and do the data support the conclusions?

Reviewer #2: Yes

3. Has the statistical analysis been performed appropriately and rigorously? 

Reviewer #2: Yes

4. Have the authors made all data underlying the findings in their manuscript fully available?

Reviewer #2: Yes

5. Is the manuscript presented in an intelligible fashion and written in standard English?

Reviewer #2: Yes

6. Review Comments to the Author

Reviewer #2: As noted in my first review, this is a good manuscript publishing. It touches on an important issue particularly for audiences interested in fertility and family planning in the DRC and that particular subregion of Africa. The authors have implemented the comments I posed during my first review. I have some additional comments and suggestions which may help to further improve the quality of the submitted manuscript, please see below.

Introduction

I think the introduction could be shortened by quite a bit.

Please provide a reference for the DRC policy stated in Line 63 through 66.

Please provide a reference for the mCPR figures reported in Line 71.

In terms of structure, I believe it is best to begin the manuscript from a broad global perspective and then narrow it down to focus on DRC. As written the manuscript begins by talking about the specific goals of the DRC in the context of its African subregion. Then in the third paragraph (Line 75 through 84) the authors suddenly move back to the global objectives regarding fertility and family planning. Much of that information in that paragraph could also be condensed to a few lines. In the interest of keeping the introduction a bit shorter, I think the authors can reference those global mobilization efforts and mention that they have influenced the DRC’s specific policies, as opposed to devoting one paragraph to them.

It is best to provide the full name of the meeting as opposed to simply describing it as the London Summit (Line 76)

I believe the authors can also shorten the two paragraphs that situate the idea of access (Line 86 - 91) and measuring access (Line 93 through 109). Perhaps those two paragraphs could be consolidated as well. For example, in the paragraph that discusses measuring access and the tools available; I feel the conversation abut the limitations of DHS and SPA data can be saved for the discussion. Simply focus the paragraph on the advantage of using

The last paragraph (Line 128 - 138) is not necessary, especially after the manuscript’s broad objectives and aims have stated previously.

Methods

It is a good idea to have a couple of sentences to explain the choices of the constructs in the methodological framework perhaps tying it back to the idea of an inadequate definition of access and the difficulties associated with measuring access. I am asking for some justification in choosing those six constructs (Line 144). Also, the placement of that paragraph seems odd, given that it has no connection to the measurements section of the paper which is what the conceptual framework serves.

I am not sure the data subsection should be used for contrasting DHS and SPA surveys with the survey design of PMA2020. Really this subsection should focus on describing PMA2020 and its design. It may also be relevant to state or highlight the partnership that led to the PMA2020 surveys.

Results

The grammar is a bit muddled in the initial reports of findings. The authors report certain figures as if they were looking at cross-sectional data as opposed to multiple waves of data. For example, Line 333, the authors mention that “Women had 1.7-1.8 live children on average at the time of the interview”. When I think they mean to say that across the four waves, average live child ranged from 1.7 to 1.8. Be clear that those statistics reported range over the different rounds of PMA2020.

The heading of Table 02 should indicate that it contains bivariate analysis results.

Pick between the adjectives, “highly” or “hugely” in Line 344.

The authors need to think a bit about what figures are reported versus those that are not for example. In Lines 357 and 358 they report the mean number of FP methods offered during each wave. They choose to report it as a range. However, given the mean methods available peaks in round 3 and then declines it may be better to just note which survey round had the highest mean of modern methods, as a precursor to a discussion.

Please carefully check the use of acronyms, as LAPM is sometimes written as LAMP (see Lines 360 to 362 as an example).

In reporting figures, there is no consistency in the use of decimal places. For example, Table 03 uses one decimal place for all the figures. While Table 04 has some p-values reported with four decimal places. Do check with the journal, typically p-values less than 0.001 are simply reported as <0.001. Review the journal's guidelines and be consistent in the reporting of significant figures.

Discussion

The first few sentences in the paragraph beginning on Line 536 and going through Line 539 are confusing. Why not just state that enumeration areas do not equate to actual communities and that in reality, communities cross the demarcations produced in establishing EAs.

I think this section of the manuscript is where you bring up the shortcomings of DHS data which is what has typically been used for this type of analysis and how by using PMA2020 data you were able to avoid those shortcomings. As opposed to discussing those issues extensively within the introduction, as is currently the case.

Conclusion

Please review the paragraph beginning with Line 592. It is not entirely clear what is being proposed as a policy intervention to affect the supply chain of LAPMs.

Additionally, the idea of FP supply is introduced rather late in the manuscript. Is this something the authors came across when researching the extant literature? Alternatively, did they find instances of other countries on the region/subcontinent struggling with supply as a policy issue? It may be something to bring up in the introduction then.

7. PLOS authors have the option to publish the peer review history of their article (what does this mean?). If published, this will include your full peer review and any attached files.

Reviewer #2: No

---

## [Author Response · Author response to Decision Letter 1]

25 Jun 2020

We appreciate the reviewer’s comments to strengthen the presentation of this study. We have responded to the comments and have made the changes as follows: 

Introduction

1- I think the introduction could be shortened by quite a bit.

- We have revised the introduction to shorten it. 

2- Please provide a reference for the DRC policy stated in Line 63 through 66.

- Thank you for noticing. We added the citation.

3- Please provide a reference for the mCPR figures reported in Line 71.

- We added the citation.

4- In terms of structure, I believe it is best to begin the manuscript from a broad global perspective and then narrow it down to focus on DRC. As written the manuscript begins by talking about the specific goals of the DRC in the context of its African subregion. Then in the third paragraph (Line 75 through 84) the authors suddenly move back to the global objectives regarding fertility and family planning. Much of that information in that paragraph could also be condensed to a few lines. In the interest of keeping the introduction a bit shorter, I think the authors can reference those global mobilization efforts and mention that they have influenced the DRC’s specific policies, as opposed to devoting one paragraph to them.

- We revised the introduction to follow the recommended order. 

5- It is best to provide the full name of the meeting as opposed to simply describing it as the London Summit (Line 76)

- We edited the name. However, we decided to delete that sentence to shorten the introduction.

6- I believe the authors can also shorten the two paragraphs that situate the idea of access (Line 86 - 91) and measuring access (Line 93 through 109). Perhaps those two paragraphs could be consolidated as well. For example, in the paragraph that discusses measuring access and the tools available; 

- We have revised the mentioned paragraph to be more concise. 

7- I feel the conversation abut the limitations of DHS and SPA data can be saved for the discussion. Simply focus the paragraph on the advantage of using

- We believe that the difference between the previous instruments and surveys to measure access and PMA2020 is important. However, in our discussion we tried to emphasize the significance of the results of current study in the context of the DRC rather than discussing the survey. In addition, Choi et al. have published an article using DHS to measure access and have discussed the shortcomings of the instrument.

8- The last paragraph (Line 128 - 138) is not necessary, especially after the manuscript’s broad objectives and aims have stated previously.

- We deleted the paragraph. 

Methods

9- It is a good idea to have a couple of sentences to explain the choices of the constructs in the methodological framework perhaps tying it back to the idea of an inadequate definition of access and the difficulties associated with measuring access. I am asking for some justification in choosing those six constructs (Line 144). Also, the placement of that paragraph seems odd, given that it has no connection to the measurements section of the paper which is what the conceptual framework serves.

- We have adopted the framework from previous relevant publications on frameworks of access to family planning service (Bertran et al. 1994 and Choi et al. 2016)

10- I am not sure the data subsection should be used for contrasting DHS and SPA surveys with the survey design of PMA2020. Really this subsection should focus on describing PMA2020 and its design. It may also be relevant to state or highlight the partnership that led to the PMA2020 surveys.

- We edited the paragraph to follow your recommendation.

Results

11- The grammar is a bit muddled in the initial reports of findings. The authors report certain figures as if they were looking at cross-sectional data as opposed to multiple waves of data. For example, Line 333, the authors mention that “Women had 1.7-1.8 live children on average at the time of the interview”. When I think they mean to say that across the four waves, average live child ranged from 1.7 to 1.8. Be clear that those statistics reported range over the different rounds of PMA2020.

- We revised the result section to clarify.

12- The heading of Table 02 should indicate that it contains bivariate analysis results.

- Table 2 shows the results based on Analysis of Variances and Chi Square test between different round. We clarified this in the text and the explanation under the table.

13- Pick between the adjectives, “highly” or “hugely” in Line 344.

- We appreciate the reviewer picking this typo.

- 

14- The authors need to think a bit about what figures are reported versus those that are not for example. In Lines 357 and 358 they report the mean number of FP methods offered during each wave. They choose to report it as a range. However, given the mean methods available peaks in round 3 and then declines it may be better to just note which survey round had the highest mean of modern methods, as a precursor to a discussion.

- We appreciate your recommendation. We revised the result section to follow your recommendation.

- 

15- Please carefully check the use of acronyms, as LAPM is sometimes written as LAMP (see Lines 360 to 362 as an example).

- We appreciate you noticing this typo. We corrected the error across the text.

16- In reporting figures, there is no consistency in the use of decimal places. For example, Table 03 uses one decimal place for all the figures. While Table 04 has some p-values reported with four decimal places. Do check with the journal, typically p-values less than 0.001 are simply reported as <0.001. Review the journal's guidelines and be consistent in the reporting of significant figures

- We appreciate the reviewer for catching this problem. We fixed the inconsistencies in Table 3.

Discussion

17- The first few sentences in the paragraph beginning on Line 536 and going through Line 539 are confusing. Why not just state that enumeration areas do not equate to actual communities and that in reality, communities cross the demarcations produced in establishing EAs.

- We followed the reviewer’s directions to clarify the mentioned limitation of this study.

18- I think this section of the manuscript is where you bring up the shortcomings of DHS data which is what has typically been used for this type of analysis and how by using PMA2020 data you were able to avoid those shortcomings. As opposed to discussing those issues extensively within the introduction, as is currently the case.

- We appreciate the reviewer’s recommendation. We agree on this point that the difference of DHS and PMA2020 methodology and instruments are important when we measure access using these resources. However, in the current study aimed to practically usePMA2020 results in the context of the DRC rather than discussing the survey. In addition, Choi et al. have published an article using DHS to measure access and have discussed the shortcomings of the instrument.

Conclusion

19- Please review the paragraph beginning with Line 592. It is not entirely clear what is being proposed as a policy intervention to affect the supply chain of LAPMs.

- We revised the conclusion to clarify.

20- Additionally, the idea of FP supply is introduced rather late in the manuscript. Is this something the authors came across when researching the extant literature? Alternatively, did they find instances of other countries on the region/subcontinent struggling with supply as a policy issue? It may be something to bring up in the introduction then.

-We revised the conclusion to clarify.

---

## [Editor Report · Decision Letter 2]

29 Jun 2020

The influence of health facility-level access measures on modern contraceptive use in Kinshasa, DRC

PONE-D-19-26324R2

Dear Dr. Babazadeh,

We’re pleased to inform you that your manuscript has been judged scientifically suitable for publication and will be formally accepted for publication once it meets all outstanding technical requirements.

Kind regards,

Kannan Navaneetham, PhD

Academic Editor

PLOS ONE
---

## [Editor Report · Acceptance letter]

9 Jul 2020

PONE-D-19-26324R2 

The influence of health facility-level access measures on modern contraceptive use in Kinshasa, DRC 

Dear Dr. Babazadeh:

I'm pleased to inform you that your manuscript has been deemed suitable for publication in PLOS ONE. Congratulations! Your manuscript is now with our production department. 

Kind regards, 

on behalf of

Professor Kannan Navaneetham 

Academic Editor

PLOS ONE